# Influence of Mechanical Loading on the Process of Tribochemical Action on Physicochemical and Biopharmaceutical Properties of Substances, Using Lacosamide as an Example: From Micronisation to Mechanical Activation

**DOI:** 10.3390/pharmaceutics16060798

**Published:** 2024-06-13

**Authors:** Elena V. Uspenskaya, Ekaterina Kuzmina, Hoang Thi Ngoc Quynh, Maria A. Komkova, Ilaha V. Kazimova, Aleksey A. Timofeev

**Affiliations:** 1Department of Pharmaceutical and Toxicological Chemistry, Medical Institute, Peoples’ Friendship University of Russia Named after Patrice Lumumba (RUDN University), 6 Miklukho-Maklaya Street, Moscow 117198, Russia; kkuz11@inbox.ru (E.K.); 1042225093@rudn.ru (H.T.N.Q.); mariya.komkova97@mail.ru (M.A.K.); kazimova.ilaha96@gmail.com (I.V.K.); 2Scientific and Educational Resource Centre “Innovative Technologies of Immunophenotyping, Digital Spatial Profiling and Ultrastructural Analysis”, Peoples’ Friendship University of Russia Named after Patrice Lumumba (RUDN University), 6 Miklukho-Maklaya Street, Moscow 117198, Russia; alexpismo77@mail.ru

**Keywords:** mechanical loading, antiepileptic drug, particle size, shape and morphology, stress field, dispersity phenomenon, solubility/dissolution rate, Spirotox, comparative dissolution kinetics test

## Abstract

Many physical and chemical properties of solids, such as strength, plasticity, dispersibility, solubility and dissolution are determined by defects in the crystal structure. The aim of this work is to study in situ dynamic, dispersion, chemical, biological and surface properties of lacosamide powder after a complete cycle of mechanical loading by laser scattering, electron microscopy, FR-IR and biopharmaceutical approaches. The SLS method demonstrated the spontaneous tendency toward surface-energy reduction due to aggregation during micronisation. DLS analysis showed conformational changes of colloidal particles as supramolecular complexes depending on the loading time on the solid. SEM analysis demonstrated the conglomeration of needle-like lacosamide particles after 60 min of milling time and the transition to a glassy state with isotropy of properties by the end of the tribochemistry cycle. The following dynamic properties of lacosamide were established: elastic and plastic deformation boundaries, region of inhomogeneous deformation and fracture point. The ratio of dissolution-rate constants in water of samples before and after a full cycle of loading was 2.4. The lacosamide sample, which underwent a full cycle of mechanical loading, showed improved kinetics of API release via analysis of dissolution profiles in 0.1 M HCl medium. The observed activation-energy values of the cell-death biosensor process in aqueous solutions of the lacosamide samples before and after the complete tribochemical cycle were 207 kJmol^−1^ and 145 kJmol^−1^, respectively. The equilibrium time of dissolution and activation of cell-biosensor death corresponding to 20 min of mechanical loading on a solid was determined. The current study may have important practical significance for the transformation and management of the properties of drug substances in solid form and in solutions and for increasing the strength of drug matrices by pre-strain hardening via structural rearrangements during mechanical loading.

## 1. Introduction

This article considers the use of tribochemical technologies, the results of which can form the basis for modifying the production process of pharmaceutical substances and obtaining a product with desired properties.

Tribochemical technologies (from the Greek *τρίβω*—to rub) are based on the processes of transferring the energy of high-intensity mechanical action to a solid body and studying the properties of substances that have undergone local, submicroscopic enrichment with energy. In this case, centres with increased activity are formed on the newly formed solid surfaces (microcomposites). The resulting stress fields relax through certain channels (mechanisms) with the release of heat and excess energy. As a result, the activation energy of the chemical transformation decreases, as follows [1]:*E*_*exc*_ = *E_a_* − *ϕ_exc_*,(1)
where *E_exc_* is the excess energy stored in structural defects of the solid; *E_a_
*is the activation energy of chemical transformation in the absence of mechanical processing; *ϕ_exc_* is the fraction of *E_exc_* that affects the activation energy of a chemical reaction.

According to [2], the accumulation rate of excess energy is determined based on the following equation:(2)$$\frac{d\emptyset }{d\tau }=\kappa -\emptyset m\exp\left(-\frac{E_{a\left(rel\right)}}{RT}\right)-\frac{A}{\propto }\frac{d\alpha }{d\tau }$$
where *κ* is the accumulation rate constant *E_exc_*; *m* is the pre-exponential factor; $E_{a\left(rel\right)}$ is the relaxation activation energy *E_exc_*; *A* is the coefficient; $\frac{d\propto }{d\tau }$ is the rate of chemical transformation; α is a value indicating the ratio of the mass of the reaction product to the mass of the entire mixture; *R* is the universal gas constant; *T*(*K*).

Tribochemistry is closely intertwined with another complex concept as a mechanochemistry. Mechanochemistry is a ‘chemical transformation caused by the direct absorption of mechanical energy’ along with electrochemistry, photochemistry and reactions occurring in solutions [3]. In the interpretation of the German researcher Heinicke G., “mechanochemistry is a branch of chemistry dealing with chemical and physicochemical transformations of substances in all states of aggregation, occurring under the influence of mechanical energy” [4]. In this regard, they distinguish among the following: chemistry of contact surfaces (tribochemistry, TrbCh), grinding chemistry (triturachemistry, TrtCh) and sonic chemistry (sonochemistry, SnCh) [5]. Thus, TrbCh, as the most important component of mechanochemistry (McCh), is accompanied by chemical transformations between two solid surfaces [6]. However, the principles of “green chemistry”—ecofriendly and sustainable reactions in the absence of or with minimal use of solvents and with minimal generation of unwanted by-products—give McnCh incomparable advantages over other reaction approaches [7,8,9] (Figure 1).

Interest in mechanochemical technologies in the pharmaceutical field began to be shown only in the twentieth century, with the production of amorphous solid dispersions and the discovery of new crystal forms (cocrystals) [10]. At the same time, the mechanochemical possibilities of biologically active substances were demonstrated. Such substances were associated with increases in dispersity, specific surface area, solubilization and viscosity of the resulting solutions, amorphization, and the bioavailability of the active pharmaceutical ingredient, as well as a decrease in crystallinity [11,12]. In the 21st century, mechanochemistry has undergone a rebirth due to the development of technology for creating mechanophores—structural fragments sensitive to applied mechanical energy [13]. The introduction of mechanophores contributed to the control of reactivity during mechanochemical modification or strengthening of the solid [14,15].

Many of the listed approaches in tribochemistry and mechanochemistry contribute to the creation of low-dose therapeutics with higher pharmacological activity [16]. Mechanical activation leads to a change in the structural and physicochemical properties of the solid. Surface changes lead to an increase in the surface area of particles and improved conditions for diffusion along interphase surfaces, as well as to a violation of the translational symmetry of the crystal and the accumulation of defects, based on dimensionality: [17,18,19,20] (Figure 2).

The defects listed above, one way or another, affect the structure-sensitive properties of solids (electrical, optical, photoemission, magnetic) as a result of the occurrence, in particular, of vibrationally and electronically excited states of interatomic bonds. Many of the structure-sensitive properties of solid materials are exceptional and non-trivial, arising only as a result of defects in the crystal. For such systems with a developed defect structure and accumulated excess free energy, the state of thermodynamic equilibrium is disrupted, which provides them with increased reactivity [21].

This fact is especially attractive for achieving the goals of medical McnCh: numerous examples have demonstrated an increase in therapeutic efficiency, the width of the therapeutic window, etc. for therapeutic agents after high-intensity mechanical impact [22,23,24,25,26,27].

According to the World Health Organization (WHO), epilepsy is one of the most common neurological diseases in the world. It affects about 50 million people, although up to 70% of people with epilepsy can live without seizures if properly diagnosed. According to the European Medicines Agency (EMA), an effective therapeutic agent used to treat partial-onset seizures is lacosamide (Vimpat), which belongs to the group of functionalized amino acids. Lacosamide is an antiepileptic drug used to treat seizures. As a chiral functionalized amino acid, it works by blocking slowly inactivating components of voltage-gated sodium currents. Lacosamide exhibits a stereoselective mode of interaction with sodium channels. Lacosamide was first approved by the European Commission (in August 2008) and was later approved by the FDA (in October 2008). It was granted approval by Health Canada (in September 2010) [28,29]. The mechanism of lacosamide action is suggested to be selective enhancement of the slow inactivation of voltage-gated Na^+^ channels, which stabilises excitable neuronal membranes. It has also been shown that lacosamide binds to CRMP-2, a phosphoprotein that is mainly expressed in the nervous system and is involved in neuronal differentiation and control of axonal outgrowth. The biopharmaceutical properties of lacosamide are presented in Table 1.

Despite the advances made in epileptology, drug-resistant epilepsy (DRE) accounts for about 30% of all cases of epilepsy. Pharmacological treatment for DRE often consists of polytherapy. However, treatment for DRE should optimize efficacy while anticipating the risks of side effects associated with polypharmacy [32]. As lacosamide treatment has demonstrated fewer side effects on systems compared to established treatments, there is a need for future larger and higher quality clinical trials to investigate the safety and efficacy of lacosamide in the treatment of comorbidities associated with epilepsy.

The purpose of the research was to study the in situ properties (surface, dynamic, dispersion, chemical, biological) of a modified lacosamide powder after a tribochemical cycle under mechanical loading, using a set of analytical approaches to control the pharmaceutical properties of a therapeutic substance.

## 2. Materials and Methods

This section describes in sufficient detail the objects of study and the methods and techniques (tribochemical, optical biopharmaceutical, chemical) that were used to obtain and discuss the results.

### 2.1. Powdery Material

The study was carried out on the high-purity (≥99, 9%) pharmaceutical substance lacosamide (Lcs), band names Motpoly and Vimpat) produced by the Jiangsu Aimi Tech Co., Ltd. (Changzhou, China) series number LM0010322, expiry date 1 February 2025 (Figure 3).

Appearance: white or almost white, or light yellow powder. Solubility: sparingly soluble in water, freely soluble in methanol, practically insoluble in heptane (Table 2).

### 2.2. Tribochemical Equipment

High-intensity mechanical loading (ML) on the Lcs substance was carried out using a laboratory knife mill with a Stegler LM-250 rotor-type brush motor (Shenzhen Bestman Instrument CO., Ltd., Shenzhen, China). The intensity of mechanical impact forces is ensured by the following equipment characteristics: rotation speed 28,000 rpm and power 13 kW (Figure 4).

#### Study Design

The stages of tribochemical influence on Lcs powder included the following: loading a mass of substance in the native state into a grinding container by ½ of its volume, continuous high-intensity mechanical impact—loading for 90 min with discharge of samples of substances every 10 min and measuring the temperature of the grinding container of the knife mill using a non-contact pyrometer; further study of dispersion, spectral and biopharmaceutical properties of samples in situ.

### 2.3. Optical Microscopy (OM) Method

Size and shape of Lcs samples before and after ML was carried out using a microscope with a special binocular attachment (Altami BIO 2, St. Petersburg, Russia) with magnification 10x (linear field of view 20 mm). The sample was applied to a glass slide and spread evenly over the entire surface. The preliminary calibration was carried out using a micrometre object with a scale of 1DIV = 0.01 mm. The particles were observed in separate fields of view. The length was measured on microscopic images, and the shape of the particles was determined using the Altami Studio 3.3 software system. Particles were observed in separate fields of view.

### 2.4. Scanning Electron Microscopy (SEM)

To analyse the morphology of Lcs samples before and after mechanical loading in terms of size, shape and surface texture, a fourth-generation scanning electron microscope (SEM) (LYRA3, Tescan, Brno-Kohoutovice, Czech Republic) with a Schottky cathode was used, with a maximum resolution of 1.2 nm and a maximum magnification of 1,000,000. Lcs samples were vacuum-treated and mounted on the stage in the SEM chamber on double-sided conductive carbon tape. To remove the charge leading to deterioration in the quality of micrographs, a thin amorphous carbon layer was deposited on the surface of the sample [33].

### 2.5. Fourier-Transform IR Spectroscopy

To study intramolecular changes associated with high-intensity tribochemical impact, vibrational spectra of Lcs and its loaded samples were recorded on a Cary 660 FT-IR spectrometer (Agilent Technologies, Santa Clara, CA, USA) with an ATR attachment with a diamond prism in the scanning range of 500–4000 cm^−1^ with a resolution of 4 cm^−1^. A small amount of the powder sample was placed on the surface of the diamond crystal, then pressed against the crystal to ensure uniform distribution and contact of the sensor with the surface.

### 2.6. Dynamic Light Scattering (DLS)

The size distribution of Lcs particles from 0.1 nm to 1000 nm, their electrokinetic potential and dispersion control in aqueous solutions before and after high-intensity mechanical loading were determined using a ZetasizerNano ZS dynamic light-scattering (DLS) spectrometer (MALVERN Instruments, Malvern, UK). The dynamic light-scattering method is based on the analysis of fluctuations in the intensity of light scattered by particles in a state of chaotic Brownian motion. As a result of fluctuation analysis, the diffusion coefficient was determined and the hydrodynamic radius of particles was calculated on the basis of the Stokes-Einstein equation: the Stokes–Einstein equation:(3)$$D=\frac{k_{B}T}{{6\pi \mu r}_{H}}$$
where *D* is the diffusion coefficient (in the case of spherical particles), *μ* (Pa.s) is the dynamic viscosity of the medium, and *r* (mol) is the radius, which can be derived via the molar volume and Avogadro’s constant. *T*(*K*) is the absolute temperature, and *k_B_* = 1.3806 × 10^−23^ J·molecule^−1^K^−1^ is the Boltzmann constant.

The DLS method was used to determine the most important quantitative characteristics of Lcs nanodispersions: Z-average size, polydispersity index (PdI), as a measure of sample heterogeneity as a result of particle aggregation and zeta potential (ZP), as a measure of the aggregative stability of colloids.

### 2.7. Static Light Scattering (SLS)

To determine the particle-size distribution (“size spectrum”) from 1 to 180 μm, we used the laser-diffraction method (low-angle laser light scattering, LALLS, Malvern Instruments, Malvern, UK), which is based on recording the scattering indicatrix that arises during the interaction of electromagnetic radiation with particles of the dispersed phase, the sizes of which were commensurate with or less than the wavelength (according to the Mie scattering). The resulting light-scattering pattern is represented by a characteristic ring structure that shows the diffraction of light waves [34].

*n*-Hexane (ACS Reagent, for organic synthesis, prep-LC, and general laboratory use, >99.9%, Merck, Rahway, NJ, USA) was used as a background and as a medium for preparing a heterogeneous solution. The SLS method was used to determine the integral dispersion characteristics of the studied samples: laser scattering, volume concentration (VC, %), specific surface area (ssa, m^2^/cm^3^).

### 2.8. Spirotox-Method Study Design

The study of the biological activity/toxicity of Lcs samples was carried out using the Spirotox bioassay method [35]. For this purpose, ciliated protozoan *Sp. ambigua* (3–5 adults) were placed in a thermostated cell filled with 0.5% aqueous solution of the Lcs test substance. Observation of the behavioral response and fixation of the time of cell death were recorded by successive signs: convulsions-twisting-cessation of motor activity. The temperature (Arrhenius) dependence of the lifetime of *Sp. ambigua* was studied (T = 297–305 K) with subsequent calculation of the activation energy of the process of cellular transformations described by the schematic diagram (Figure 5).

### 2.9. Dissolution Rate Kinetics

The SLS (LALLS) method was also used to study the kinetics of dissolution of the studied Lcs samples in water according to a previously developed method [36]. To do this, a sample of Lcs substance powder of about 0.03 g was placed in a cell with 3 mL of water. Measurements were continued at intervals t = 10 s. The moment of completion of the measurement was recorded by the cessation of change in time of the Laser Obscuration (LO) value, which characterizes the loss of light intensity when introducing a dispersed sample into the measuring cell (formula (4)). The results are presented in the LO-t,sec coordinates. The dissolution rate constant was calculated from the slope of the linearized section of the straight line to the abscissa axis in semi-log coordinates:(4)$$Laser\;Obscuration\;(LO)=1-\frac{I}{I_{0}}\times 100\%$$
(5)k=−tg⁡α
where *I* is the light intensity measured by the detector in the presence of a sample in the cell, *I*_0_ is the light intensity measured by the detector in the absence of a sample, *k* is the dissolution rate constant, s^−1^.

### 2.10. Comparative Dissolution Kinetics Test (CDKT)

We used the comparative dissolution kinetics test as a prognostic tool, including a comparison of the dissolution profiles of unloaded and loaded samples (t_ML_ = 90 min). The studies were carried out using a dissolution tester (model UTD812A, Logan Instruments Corp., Somerset, NJ, USA) with the paddles apparatus (USP II). The selection of sampling points, as well as that of dissolution conditions, was carried out in accordance with the recommendations of the FDA for conducting CDKT (U.S. Food and Drug Administration) [37]. The CDKT conditions were as follows: dissolution medium—0.1 M hydrochloric acid solution; volume of dissolution medium—900 mL; temperature of dissolution media—37 ± 0.5 °C; stirrer rotation speed—20 ± 2 rpm; time points—5; 10; 15; 20; 30 and 40 min.

The CDKT design included the following stages: a sample of the substance was placed into a dissolution beaker and dissolved. At the indicated time intervals, aliquots of 10 mL were taken with a mechanical pipette, replenishing the taken volume with dissolution medium, and filtered through a membrane filter (GS-Tek SN02545 Nylon, Newark, DE, USA) with a pore diameter of 0.45 μm, discarding the first portion of the filtrate. The aliquots taken were used for subsequent analysis on a high-performance liquid chromatograph equipped with a spectrophotometric detector, model SPD-20A (Shimadzu, Kyoto, Japan). The analysis used a ZORBAX Eclipse XDB chromatography column with a particle size of 5 μm, 150 mm × 4.6 mm, (Agilent, Santa Clara, CA, USA), mobile phase of acetonitrile for chromatography/water for chromatography in a volume ratio of 13/87, with a flow rate of 2 μL/min and injection volume of 5 µL. Lacosamide detection was carried out at a wavelength of 215 nm.

The concentration of lacosamide that went into solution (X, %) was calculated using the following formula:(6)$$X=\frac{S\times a_{0}\times V\times P\times 100}{S_{0}\times 100\times a\times 100}=\frac{S\times a_{0}\times P\times V}{S_{0}\times a\times 100}$$
where, *S* is the lacosamide peak area in the chromatogram of the test solution; *S*_0_ is the average value of the lacosamide peak area at *n* = 5 on chromatograms of the standard solution; *a*_0_ is the weighed portion of the standard sample of lacosamide, (mg); *V* is the volume of dissolution medium, (mL); *P* is the API content (%) in the reference lacosamide sample; and a is the weighed portion of the tested substance (mg).

For each subsequent time point, the dilution factor of the solution after sampling was taken into account:(7)$$X=\left(\frac{S_{i-1}\times 10}{900}+S_{i}\right)\times \frac{a_{0}\times V\times P}{S_{0}\times 100\times L}$$
where, *S_i_*_−1_ is the area of the lacosamide peak in the chromatogram of the test solution at the previous time point; *S_i_* is the peak area of lacosamide in the chromatogram of the test solution at the time point under study.

### 2.11. Statistical Analysis

Data were reported as mean ± SD, using the unpaired Student’s *t*-test. Values of * *p* < 0.05 and *** *p* < 0.001 were considered significant and extremely significant, respectively.

## 3. Results and Discussion

The hierarchy of the structure of a solid body that has undergone a high-intensity impact in the form of mechanical loading, as well as the need to identify new, previously unknown features of the structure, drives the study of deformation processes based on a multi-level approach (macro-, meso- and micro-scales) [38,39].

### 3.1. Particle Size Distributions (PSDs)

To analyses the dispersibility of aqueous and non-aqueous solutions and the effect of Trb impact on the studied powdery substances, laser-based analysis methods were used based on the monochromatic, coherent, narrowly directed radiation flux interaction with particles, their ensembles or fluctuations scattering.

#### 3.1.1. Static Light Scattering

Figure 6 shows the “size spectra” of the distribution of ensembles of particles in the range from 1 to 120 μm, as well as the integral characteristics of the dispersity of powdered Lcs in the native sample and after the Trb impact.

The figure shows the dispersity phenomenon (DPh) previously described in [40,41,42,43,44]. In this study, the DPh phenomenon is observed in the increase in diameter (d, μm) and volume fraction (%) of size groups of Lcs particles in the ongoing micronisation process at high-intensity ML applied to the solid. A drop in specific surface area values characterizes a decrease in the dispersion degree of the studied samples that reaches its minimum at t = 30 min of high-intensity ML (see Figure 6b). Further changes in ssa values are in a state of non-significant fluctuations in the region of low values.

The propagation of stress waves preceding the DPh phenomenon leads to a defective crystal structure, accumulation of excess free surface energy Fs of the dispersion system and the tendency to spontaneously reduce this parameter due to the aggregation of solid powder particles, and decrease in the surface area in the Trb impact process [45]:(8)$$F_{s}=\frac{dF}{dS}S=\sigma S$$
where *F_s_* is free surface energy; *σ* is free surface energy per unit surface area; *S* is the area of the interface.

Thus, the micronisation of Lcs particles is accompanied by deformation and aggregation of crystals, amorphization and formation of centres with increased activity, which can result in increased reactivity [46].

#### 3.1.2. Dynamic Light Scattering

The effect of high-intensity ML lacosamide powder on the colloidal properties of its aqueous solutions was studied using the example of changes in such physicochemical properties as the hydrodynamic radius of particles (d, nm), average count rate (kcps), electrophoretic mobility (ξ, mV) and polydispersity index (PDI) (Figure 7).

As the results show, dynamic light scattering makes it possible to analyse changes in the conformation of colloidal particles of the dispersed state in Lcs solutions as a function of ML time: the largest changes in the average particle diameter, polydispersity index and, as a consequence, an increase in the average count rate and changes in zeta potential occur in first 30–50 min of ML. This fact may indicate the swelling of nanodispersed bicontinuous particles in the aqueous solution of Lcs [47]. An increase in the impact time Trb (milling time) further leads to a decrease in the average count rate (see Figure 7b): continued ML on a solid body changes the field of defects, described by the defect density tensor and defect flux density [48,49]. Changes in the intramolecular thermal motion and conformation of macromolecules lead to inhomogeneities in the refractive index of particles and fluctuations in the intensity of scattered light [50].

### 3.2. Fourier Transform Infrared (FT-IR)

Since ML of a solid is accompanied by a local increase in temperature and pressure (formation of triboplasma), phase transformations, breaking of chemical bonds, and emission of light and electrons, the emergence of vibrationally and electronically excited states of interatomic bonds entails changes in the position of the characteristic bands and their intensity. High intensity ML can lead to oxidation and, in some cases, degradation of the compound [51].

Figure 8 shows FT-IR spectra in various formats (transmission mode, T(%), subtraction of the spectra of ML samples from the native lacosamide and signal-to-noise ratio mode), allowing a significant expansion the possibilities of analysing the resulting differential spectra.

In the spectra of the loaded samples, changes in the intensity of the transmission bands compared with the native lacosamide (black colour) are clearly visible: in most cases, the curves of the loaded samples lie below the curve of the native Lcs and, therefore, are characterized by greater absorption caused by group vibrations: O-H, N-H, CO_2_, C-H. The changes are especially noticeable for samples that were exposed to mechanical loading for 30–40 min, 80–90 min.

However, only special chemometric capabilities made it possible to detect wavenumber regions corresponding to significant changes in the FT-IR spectra (see Figure 8b,c): 3500–4000 cm^−1^, 2250–1750 cm^−1^, 750–500 cm^−1^ (Table 3).

### 3.3. Surface and Near-Surface Morphology

#### Scanning Electron Microscopy (SEM)

To identify changes in the grain structure of Lcs samples at different stages of high-intensity ML, the nature of destruction, and the presence of defects, the SEM method was used. The studies were carried out in the mode of recording secondary electrons at the accelerating voltage of 10 kV and working distance of 15 mm at magnifications up to ×40,000. The results of the electron microscope studies are shown in Figure 9 and Appendix A.

Particles of the native Lcs substance are needle-shaped drusen with rough surfaces and jagged edges, ranging in size from 30 µm to 50 µm (see Figure 8a). After 60 min of loading a solid body, destruction of the Lcs sample was observed. It was accompanied by the appearance of smaller particles (d~10 μm) with brittle surfaces and sharp edges (see Figure 8b). The beginning of conglomeration was evidenced by the accumulation of particles with varying dispersion degrees. However, the anisotropy of the microstructure was maintained (see Figure 8b). Increasing the time of mechanical loading to 90 min led to the transformation of the sample into a glassy (amorphous, metastable) state with a smooth, continuous surface with isotropic properties (see Figure 8c). The morphology of the microstructure of the solid at the final stage of mechanical loading on Lcs convincingly deomonstrated the result of plastic deformation with the formation of structures characterized by linear, surface and volumetric defects [52] (see Figure 2).

### 3.4. Lacosamide Dissolution Studies

According to [53,54], the appearance of disordered regions in API crystals as a result of mechanical loading, accompanied by the growth of dislocations, leads to a change in some important pharmaceutical properties, for example, to an increase in the observed dissolution rate constant (K′^obs^) in bulk dissolution studies. In this regard, the next stage in this study of the physico-chemical properties of the Lcs substance powder was experiments in laser assessment of the kinetics of dissolution in water and the CDKT results.

#### 3.4.1. Dissolution Kinetics by LALLS

Studies of dissolution kinetics using an original approach based on the time variation of the laser light-scattering indicatrix (see Equation (4)) were carried out for Lcs samples with different ML times (Figure 10).

The figure shows an exponential decrease over time in the values of the dependent variable. Laser obscuration, however, was associated with a noticeably longer duration in the case of the native sample (black) compared to samples at t_MS_ = 30 min (green) and t_MS_ = 90 min (blue). It can be seen that the substance of the amorphous sample (t_ML_ = 90 min) dissolves noticeably faster, showing similarity of the isotropic structure with the molecular structure of the liquid [55]. The dissolution rate (k∙10^2^, s^−1^) measured by the coefficient b of the straight line equation y = a + bx in coordinates ln(1 − *I*/*I*_0_) − *t_D_* (s) (see Figure 10b), is presented in Table 4.

It is known that the dissolution rate increases with an increase in the number of dislocations, which was demonstrated by the results obtained in Table 4 [56]. According to [57], an approximately three-fold increase in the average dislocation density leads to an increase in ^obs^k(s^−1^) by 21%. Based on the presented tabular data, an approximately 20-fold increase in the dislocation density is observed by the final time of mechanical loading on lacosamide (t_ML_ = 90 min).

The effect of dislocation density on the resistance to deformation is described by the typical “stress–strain” curve [58]. An increase in the number of dislocations is also facilitated by a rise in temperature inside the grinding container of the knife mill during ML on the Lcs powder. If the temperature change inside the TrbCh reactor is considered as the response of the dispersion system to the mechanical impact that is produced and accompanied by an increase in SB deformation, it is similar to the typical “stress–strain” curve (Figure 11).

By analogy with the stress–strain curve, the shape of the temperature curve allows for predictive evaluation of dynamic changes in the SB properties, such as the tensile strength under ML and the angular coefficient of the straight section to the OX axis: k = tgα = 2.64 min^−1^ by analogy with the calculation of Young’s modulus (see Figure 11b). It can be observed that the limit of elastic deformation of lacosamide corresponds to the mechanical loading (ML) time t_Ml_ = 20 min. At t_ML_ > 20 min, the system enters the region of plastic deformation, and this is accompanied by an increase in resistance to ML and hardening of the solid body. At t_ML_ = 70 min, the system reaches its strength limit, the maximum uniform plastic deformation. At t_ML_ > 70 min, the Lcs sample undergoes uneven deformation and reaches the failure point at t_ML_ = 90 min. It is noteworthy that, according to the FT-IR method, significant changes in quantum transitions between vibrational energy levels of lacosamide are observed at t_MS_ = 80–90 min (see Figure 8d).

In general, based on the appearance of the stress–strain curve, we can conclude that the lacosamide substance is a strong but not plastic material: it stretches very little and suddenly breaks.

The study of the dissolution of Lcs samples showed the peculiarity of its aqueous solutions in the observed dependence of the dissolution parameters—time and speed, as well as the activation energy (Ea, kJmol^−1^)—of cell transformations [59] (see Figure 5) on the time of mechanical loading (ML) of the solid: the inflection point at t_ML_~20 min corresponds to the equilibrium time ML (t_eqv_), as well as to the equilibrium dissolution (d_eqv_) of the substance in water and activation of the process of cell biosensor death in aqueous solutions of Lcs (Figure 12).

It is noteworthy that the detected t_eqv_(min) corresponds to the first amplitude jump in the 2D diagram, demonstrating the change in the integral dispersion characteristics of Lcs depending on the ML time (see Figure 6b).

#### 3.4.2. In Vitro Equivalence Dissolution Test (CDKT)

According to [60], when testing the dissolution of a therapeutic agent in vitro, any change in the properties of the pharmaceutical substance (for example, particle size) must be taken into account. A comparative test of dissolution kinetics makes it possible to identify fundamental changes in the composition and properties of a therapeutic agent at stages of the production cycle that affect the rate and proportion of its release into the dissolution medium [61]. Figure 13 shows the dissolution profiles of samples of the Lcs substance at the final stage of tribochemical impact (mechanical loading, t_ML_ = 90 min) and those of the native substance, demonstrating the proportion of API released into the solution of 0.1 M HCl (pH 1.2), which simulates gastric fluid (without enzyme).

The reliability of the analysis results was confirmed using relative standard deviation (RSD) calculations at each time point (Table 5).

Both dissolution profiles demonstrate complete release of the API; that is, the curves reach a plateau with a value above 85%. However, in comparison with the native sample, the loaded lacosamide, which underwent a cycle of tribochemical impact for 90 min, demonstrates kinetic advantages: 92% is recovered into the dissolution medium in 10 min, which represents a 7% difference vs the native sample; the loaded Lcs goes completely into the dissolution medium in 15 min with 95% release, and this process is characterized by a low (>1% RSD) variability of values and exhibits a higher dissolution rate. The native (unloaded) lacosamide goes into the dissolution medium in 20 min, with a lower rate of saturated solution formation and a higher (<1% RSD) variability of values at each time point.

The approach based on the use of compression, shear and friction energy to transform a solid drug substance into an amorphous state has been widely reported in the literature [62]. Compared to the crystal form, amorphous solid pharmaceutical substances with a relatively high density of defects have significant potential and advantages for effectively improving the *per os* bioavailability of poorly water-soluble substances [63]. It is known that the lower the crystallinity of a solid, the greater the degree of disorder and, accordingly, the entropy of a given system, as, in amorphous states, there is no long-range translational order [64]. If the value of the product TΔS is large, the loss of Gibbs free energy is therefore increasing: the energy spent on the destruction of an amorphous solid is small and the increase in the enthalpy of dissolution ΔH_sol_ is mainly due to the change in enthalpy due to solvation by solvent particles ΔH_sol_. We demonstrated solvation during the swelling of Lcs colloids as a result of the penetration of solvent molecules into the nearest layers of particles of the loaded substance (see Figure 7a).

Consequently, our results based on dissolution kinetics by LALLS and an in vitro equivalence dissolution test showed a positive effect of the amorphous nature of the loaded lacosamide sample in terms of increasing the dissolution rate in different media.

## 4. Conclusions

The technology of tribochemical microphase deformation of the crystal structure of antiepileptic lacosamide was studied under conditions of mechanical loading with discharge of dispersed samples of the substance at different stages of ML. The analysis of discharged samples in situ showed that the destruction of Lcs is a complex multi-stage process at various structural levels that includes the following effects: phenomenal aggregation of particles; dispersion phenomena (macro- and meso-levels), the driving force of which was the desire of the system to reduce surface energy with a decrease in surface-area section; initiation of conglomeration with transformation into an amorphous metastable substance with isotropy of properties (microlevel) as a result of plastic deformation; changes in the position and intensity of characteristic bands in the FT-IR spectra of all unloaded samples due to the occurrence of vibrationally and electronically excited states of interatomic bonds; swelling of nanodispersed bicontinuous particles (nanolevel) in aqueous solutions; 2.4-time increase in dissolution and dissolution rates in water for samples of the substance as a result of mechanical loading; improvement of kinetic characteristics according to the CDKT method and analysis of the dissolution profile of a loaded sample in a medium simulating gastric juices; reducing the activation energy of the process of cell-biosensor death in a solution of a loaded sample as a sign of an increase in its biological activity; establishing the tensile strength according to a curve similar to the “stress–strain” curve and setting the time t_ML_~20 min as corresponding to equilibrium dissolution. Thus, micronisation of Lcs particles is accompanied by deformation and amorphization with the formation of centres with increased activity, which can lead to increased reactivity.

The conducted research may have promising practical significance in managing improved pharmaceutical properties and increasing the strength of therapeutic matrices through preliminary stress hardening due to structural changes under mechanical loading.

## Figures and Tables

**Figure 1 pharmaceutics-16-00798-f001:**
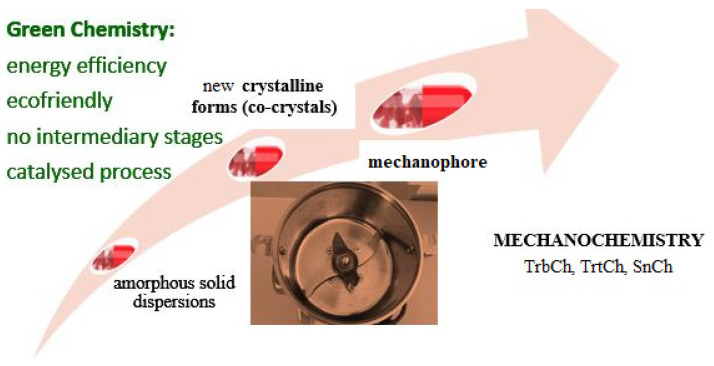
Stages and advantages of mechanochemistry.

**Figure 2 pharmaceutics-16-00798-f002:**
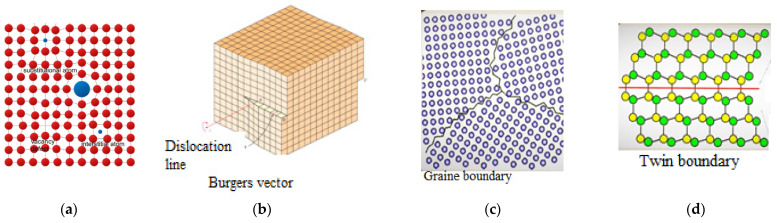
Crystal defects: (**a**) point defects (zero-dimensional); (**b**) line defects (1D); (**c**) surface defects (2D); (**d**) volume defects (3D).

**Figure 3 pharmaceutics-16-00798-f003:**
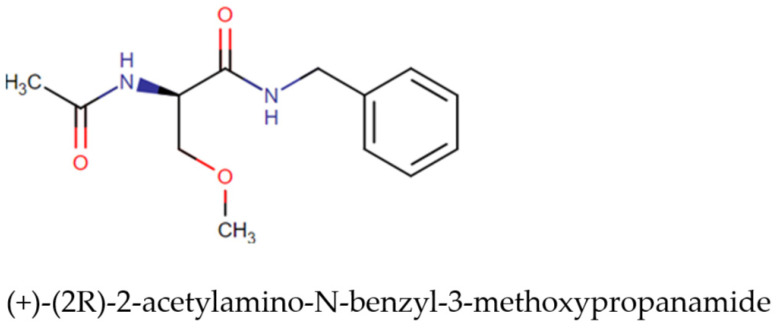
Chemical structure of lacosamide [31].

**Figure 4 pharmaceutics-16-00798-f004:**
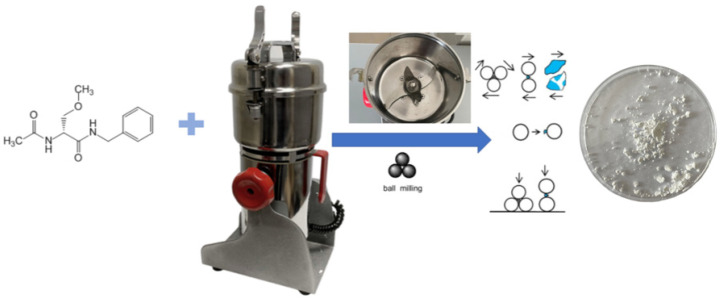
Principal schematic of the equipment for carrying out tribochemical processes.

**Figure 5 pharmaceutics-16-00798-f005:**
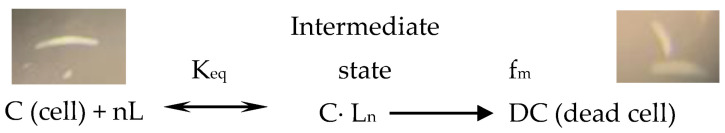
Kinetic scheme of ligand-receptor interaction *Sp. ambigua* with toxicant: C-cell, L-ligand, *n*-stoichiometric coefficient, C·L_n_—intermediate state (cell after interaction with the ligand), K_e_ is the equilibrium constant fast stage, f_m_ is the rate constant of the cell transition to the dead state, DC is a dead cell. The inserts show photographs of ciliates at the stages of incubation in the medium and recording of death.

**Figure 6 pharmaceutics-16-00798-f006:**
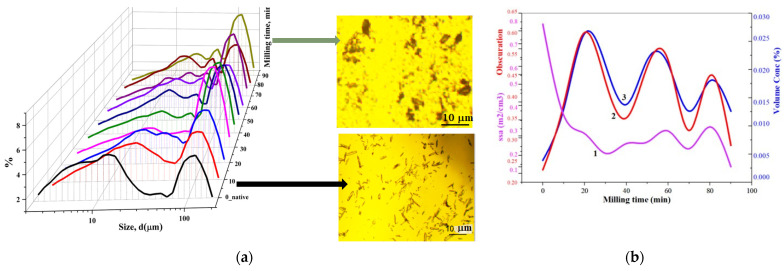
Dispersive properties of lacosamide samples at different times of Trb impact according to the SLS method: (**a**) Particle size distributions; (**b**) Integral dispersion characteristics—laser obscuration, volume concentration, V(%); specific surface area, ssa (m^2^/cm^3^). Insets show the morphology of Lcs crystals before and after high-intensity ML.

**Figure 7 pharmaceutics-16-00798-f007:**
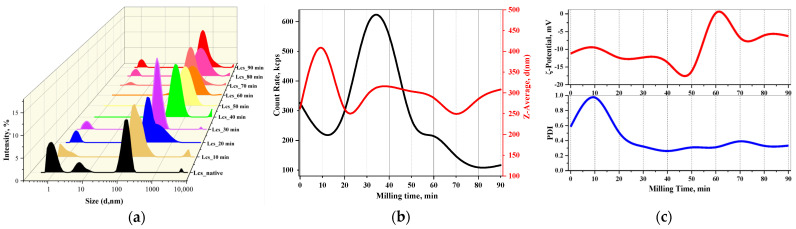
Dispersive properties of lacosamide samples under different Trb impact times according to the DLS method: (**a**) particle-size distributions in units of laser light-scattering intensity; (**b**) 2D data diagram of the average count rate and nanodispersion size; (**c**) ξ-potential (mV) and polydispersity index value (PDI).

**Figure 8 pharmaceutics-16-00798-f008:**
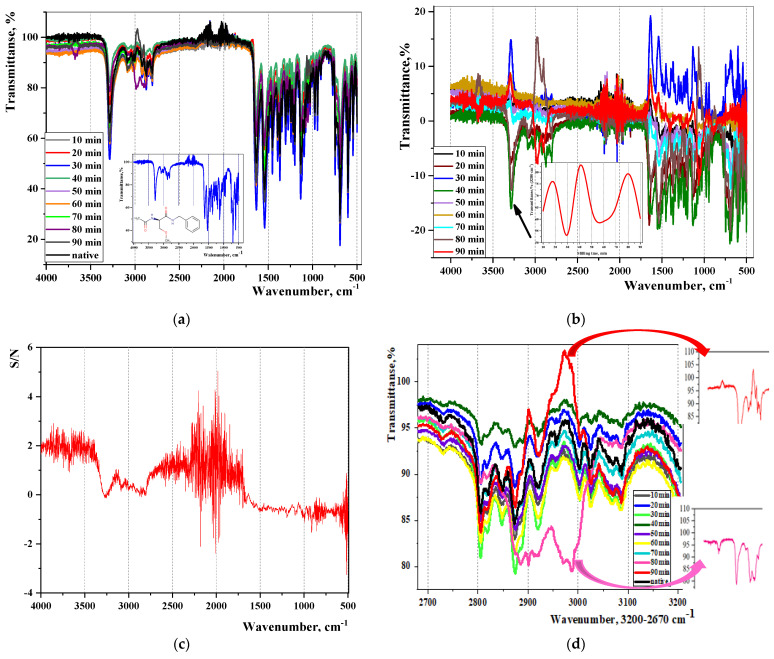
FT-IR spectra of Lcs samples before and after Trb impact: (**a**) Full range (the inset shows a naive Lcs sample); (**b**) Spectra of loaded samples “subtraction” from native Lcs (the inset shows the amplitude vibrations of the transmission maximum at 3280 cm^−1^, corresponding to the NH group); (**c**) Dependence of the signal-to-noise value on the wave number (cm^−1^); (**d**) High-frequency region (3200–2700 cm^−1^) of the range of characteristic vibrations of bonds (the insets shows the spectrum regions with reflected transmission for Lcs samples at t_ML_ = 80 min and t_ML_ = 90 min).

**Figure 9 pharmaceutics-16-00798-f009:**
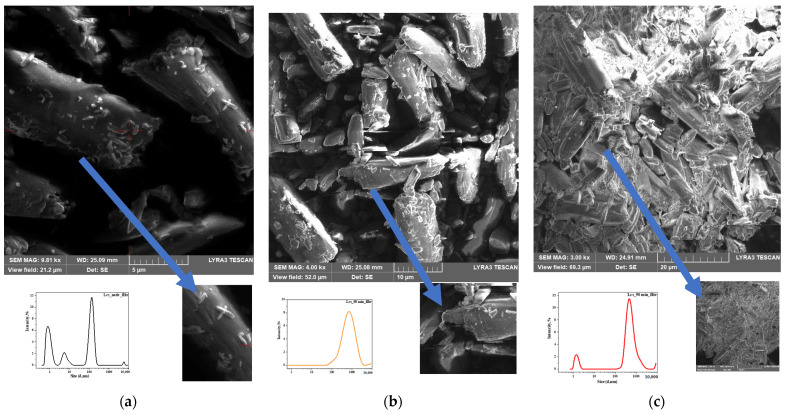
SEM micrographs for Lcs obtained at different times of high-intensity ML and: (**a**) t = 0 min; (**b**) t = 60 min; (**c**) t = 90 min. Device magnification (MAG) = 9.81–3.00 kx. The insets show the particle size distribution according to the DLS method.

**Figure 10 pharmaceutics-16-00798-f010:**
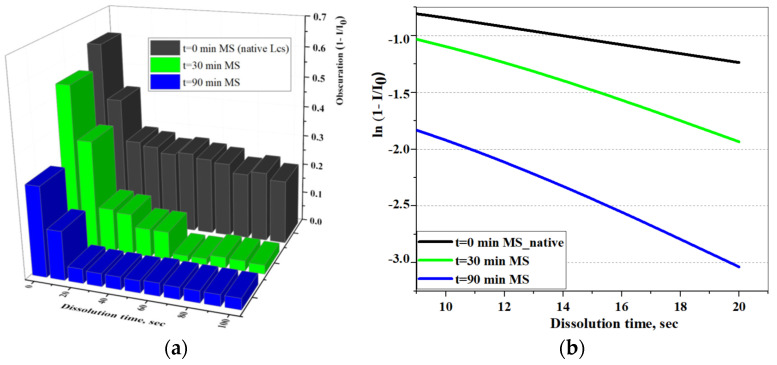
The Lsc’ samples dissolution in water as measured by LALLS method: In direct (**a**) and semi-logarithmic (**b**) coordinates.

**Figure 11 pharmaceutics-16-00798-f011:**
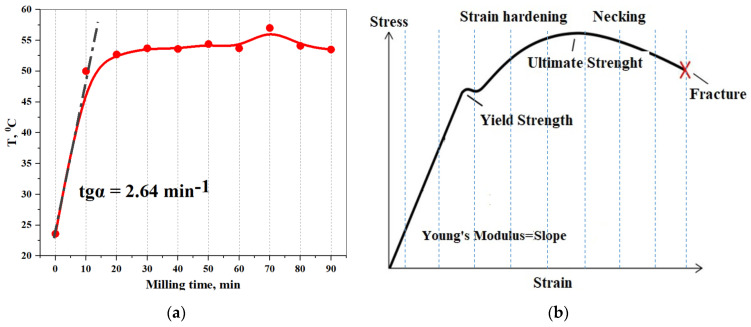
Stress–strain curve. (**a**) Temperature curve of the heating inside the milling bowl-duration of the applied ML (**b**) Typical of SB stress–strain curve.

**Figure 12 pharmaceutics-16-00798-f012:**
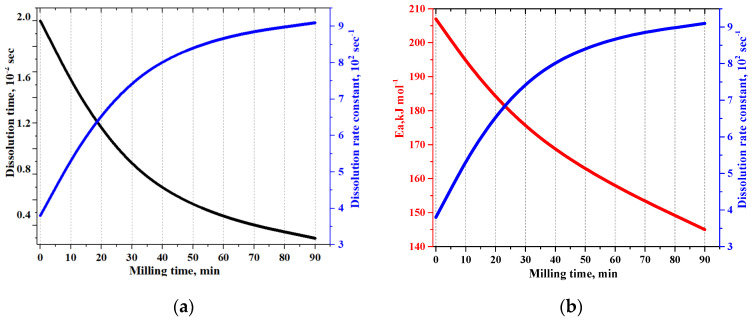
2D-diagrams showing the dependence of the parameters of dissolution in water and death of the *Spirostomum ambigua A* cell biosensor in 0.5% aqueous solutions of various ML Lcs samples: (**a**) Time and dissolution-rate constant; (**b**) activation energy of cell transformations.

**Figure 13 pharmaceutics-16-00798-f013:**
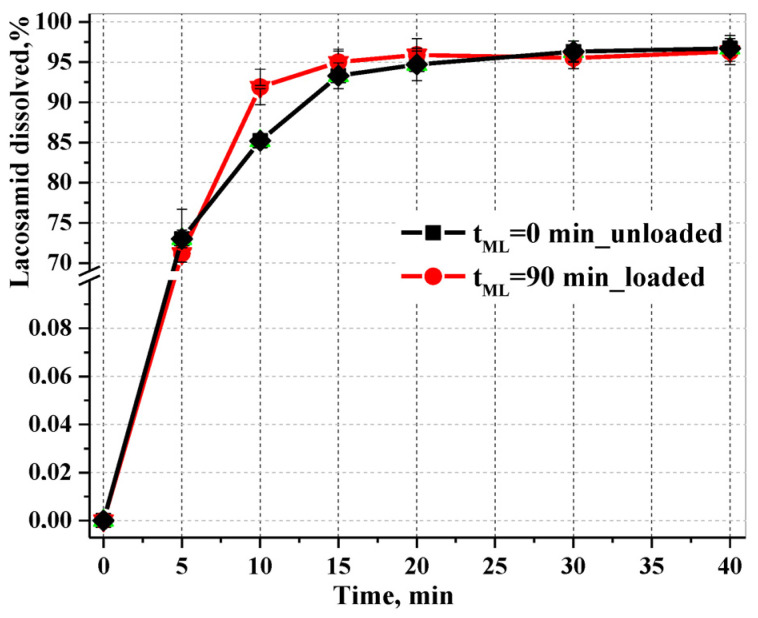
Cumulative dissolution profiles of lacosamide samples unloaded (black) and loaded (red) in simulated gastric fluid obtained using the USP II apparatus. The results are expressed as mean ± RSD (*n* = 6).

**Table 1 pharmaceutics-16-00798-t001:** Pharmacokinetic properties of lacosamide [30,31].

Absorption *, % per os	Time, h for C_max_ in Plasma	Volume of Distribution, L/kg	Bound to Plasma Proteins, %	Half-Life, h	Metabolism, % in the Urine	Reaction Type, the Major Compound	Toxicity **, per os in Rats, mg/kg
100	0.5–4	0.6	15	13	95	O-demethylation 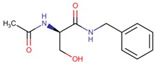	253

* for doses up to 800 mg. ** dizziness, nausea, seizures.

**Table 2 pharmaceutics-16-00798-t002:** Physicochemical properties of lacosamide.

Molecular Weight	Chemical Formula	Water Solubility, mg/mL *	log Po/w	pKa	pK_BH_+	T, melting (°C)	T, Boiling (°C)
250.3	C_13_H_18_N_2_O_3_	0.465 *	0.728 **	12.5 *	−2 *	140–146 **	536.5 **

* Predicted properties. ** Experimental properties.

**Table 3 pharmaceutics-16-00798-t003:** The main transmittance bands in the Lcs FT-IR.

Frequency Range, cm^−1^	Group	Compound Class	Appearance/ Comments
3650–3200	O-H stretching	alcohol	strong, broad/ intermolecular bonded
3350–3310	N-H stretching	secondary amine	-
3300–2500	O-H stretching	carboxylic acid	strong/usually centred on 3000 cm^−1^
3100–3000	C-H stretching	alkene	medium
2250–1800	amino acid residues, CO_2_		
1698	C=O stretching	secondary amide	strong/free associated
1690–1640	C=N stretching	imine (tautomer)	medium
1650–1580	N-H bending	amine	strong
1465	C-H bending	alkane/methylene group	medium
1450–1375	C-H bending	alkane/methyl group	medium
1420–1330	O-H bending	alcohol (tautomer)	medium
1250–1020	C-N stretching	amine	medium
1210–1163	C-O stretching	ester	strong
750 ± 20 700 ± 20	C-H bending	monosubstituted benzene derivative	strong

**Table 4 pharmaceutics-16-00798-t004:** Dissolution parameters in water of Lcs substance samples with different mechanical strain times.

Mechanical Stress Time, Min	Dissolution Time, s	Dissolution Rate Constant, k·10^2^, s^−1^ ± SD
0	200	3.80 ± 0.001
30	60	8.40 ± 0.001
90	30	9.10 ± 0.017

**Table 5 pharmaceutics-16-00798-t005:** Amount of API transferred into solution.

Time, min	Lacosamide (Unloaded) t_ML_ = 0 min		Lacosamide (Loaded) t_ML_ = 90 min	
	C, %	RSD, %	C, %	RSD, %
5	73.0	1.1	71.2	5.5
10	85.2	0.2	91.9	2.2
15	93.3	1.6	95.0	1.4
20	94.7	2.0	95.9	0.5
30	96.3	1.3	95.5	0.4
40	96.7	1.6	96.3	0.5

## Data Availability

The data presented in this study are available in this article and Appendix A.

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
