# Peer review of "Influence of Mechanical Loading on the Process of Tribochemical Action on Physicochemical and Biopharmaceutical Properties of Substances, Using Lacosamide as an Example: From Micronisation to Mechanical Activation"

_pharmaceutics, 2024, doi:10.3390/pharmaceutics16060798_

Round 1

Reviewer 1 Report

Comments and Suggestions for Authors

Dear Authors, 

The manuscript has scientific value, but it contains inaccuracies and the suggestions provided should help address them:

1. The introduction recommends a more detailed description of the properties of the active substance, such as its bioavailability. Why is it appropriate to use Lacosamid in the selected study? What are the biopharmaceutical properties of Lacosamid, how will this study help to solve biopharmaceutical problems. Describe the results of other studies in this area. This will reveal the relevance of the study.2. The materials and methods  must describe all the materials used, indicating their degree of purity, the manufacturer.3. Lines 140 to 145 contain redundant information in this section. It is better to put it in the introduction.4. Consider whether formulas are required in the results section. 5. Please review the abbreviations, their explanation must be in English. Think about it might be appropriate to give the abbreviations in the text of the article.

Author Response

Response to Reviewer 1 Comments

The authors thank the respected Reviewer for valuable comments aimed at eliminating erroneous conclusions, inaccuracies in wording and at improving the quality of the material of the article, in general, to meet the high requirements of Pharmaceutics.

Point 1: The introduction recommends a more detailed description of the properties of the active substance, such as its bioavailability. Why is it appropriate to use Lacosamid in the selected study? What are the biopharmaceutical properties of Lacosamid, how will this study help to solve biopharmaceutical problems. Describe the results of other studies in this area. This will reveal the relevance of the study.

Response 1: Thank you very much! The Manuscript’ introduction has changed:

“The mechanism of action of lacosamide is suggested to be selectively enhance the slow inactivation of voltage-gated Na+ channels, hence stabilising excitable neuronal membranes. It has also been shown that lacosamide binds to collapsin response mediator protein-2 (CRMP-2), a phosphoprotein which is mainly expressed in the nervous system and is involved in neuronal differentiation and control of axonal outgrowth. The biopharmaceutical properties of lacosamid are presented in Table 1. ”

Table 1. Pharmacokinetic properties of lacosamide [Cawello W. Clinical pharmacokinetic and pharmacodynamic profile of lacosamide. Clin Pharmacokinet. 2015 Sep;54(9):901-14. doi: 10.1007/s40262-015-0276-0.
DrugBank Online (version 5.1.12, released 2024-03-14) Available online: https://go.drugbank.com/drugs/DB06218 (accessed on 4 June 2024)]  .

Absorption*, %

per os

Time, h

for Cmax in plasma

Volume of distribution,

L/kg

Bound to plasma proteins,

 %

Half-life, h

Metabolism,

 %

in the urine

Reaction type,

the major compound

Toxicity**,

per os in rats,

mg/kg

100

0.5 -4

0.6

15

13

95

O-demethylation

253

* for doses up to 800 mg

** dizziness, nausea, seizures

«Because lacosamide treatment demonstrated a less side effects on the systems, there is a need for future larger and higher quality clinical trials to investigate both the safety and efficacy of lacosamide in the treatment of comorbidities associated with epilepsy

Point 2: The materials and methods must describe all the materials used, indicating their degree of purity, the manufacturer

Response 5: Thank you.

“The study was carried out on the high purity (≥99, 9%) pharmaceutical substance lacosamide (Lcs), band mames (Motpoly, Vimpat) produced by the Jiangsu Aimi Tech Co., Ltd. ( Jiangsu China)series number LM0010322, expiry date 1/2/2025”

n-Hexane (ACS Reagent, for organic synthesis, prep-LC, and general laboratory use, >99.9%, Merck, USA)

Point 6: Lines 140 to 145 contain redundant information in this section. It is better to put it in the introduction.

Response 6: Thank you! It has put in the Introduction section.

Point 7: Consider whether formulas are required in the results section.

Response 7: Thank you! The formula 9 has deleted.

Point 8: Please review the abbreviations, their explanation must be in English. Think about it might be appropriate to give the abbreviations in the text of the article.

Response 8: Thank you. I apologise for the omissions in translating the abbreviation into English.

The section has been corrected, amended:

Abbreviations

in situ

the original (primary, without movement) location of experiments

CRMP-2

collapsin response mediator protein-2  

DLS

dynamic light scattering

DPh

dispersity phenomenon

DSA

dynamic strain aging

DRE

drug resistant epilepsy

EMA

The European Medicines Agency

Ea

activation energy

FDA

Food and Drug Administration

FT-IR

Fourier transform IR spectroscopy

kcps

kilo counts per second

Lcs

lacosamide

LALLS

low-angle laser light scattering

MCh

mechanochemistry

MAct

mechanoactivation

ML

mechanical loading

OM

optical microscopy

PDI

polydispersion index

SnCh

sound chemistry (sonochemia)

SEM

scanning electron microscopy

SB

solid body

TrbCh

tribochemical

CDKT

Comparative Dissolution Kinetics Test (In vitro equivalence dissolution test)

WHO

World Health Organization

With gratitude, the authors

Reviewer 2 Report

Comments and Suggestions for Authors

The current manuscript discussing the effect of mechanical stress on the physicochemical and biopharmaceutical properties of lacosamide is interesting and needs to be further improved for better understanding to the readers. Please address the below comments before considering the manuscript for further processing:

1.     Abstract needs to be improved for better flow of the content. Authors are suggested to make the following aspects clear to the readers “aim, methods, results and conclusion”. Do not make it complex with formulas.

2.     Please focus discussing the objective of the current research in the introduction section of the manuscript.

3.     Is the process of knife grinding a common practice for side reduction of active pharmaceutical ingredients?

4.     Within the results, please discuss the results comparing the characteristics of lacosamide before and after milling.

5.     Please improve the discussion of dissolution studies.

Author Response

Response to Reviewer 2 Comments

The authors thank the respected Reviewer for valuable comments aimed at eliminating erroneous conclusions, inaccuracies in wording and at improving the quality of the material of the article, in general, to meet the high requirements of Pharmaceutics.

Point 1: Abstract needs to be improved for better flow of the content. Authors are suggested to make the following aspects clear to the readers “aim, methods, results and conclusion”. Do not make it complex with formulas.

Response 1: Thank you very much! The abstract has changed:

Many physical and chemical properties of solids, such as strength, plasticity, dispersibility, solubility and dissolution, are determined by defects in the crystal structure. The aim of this work is to study in situ the surface, molecular dynamic, dispersion, chemical and biological properties of modified lacosamide powder by laser scattering, electron microscopy, FR-IR and biopharmaceutical methods as a result of a complete mechanical loading cycle. The SLS method demonstrated the phenomenon of spontaneous tendency of surface energy reduction due to aggregation during micronisation. DLS analysis showed conformational changes of colloidal particles as supramolecular complexes depending on the loading time on the solid. SEM analysis demonstrates the conglomeration of needle-like lacosamide particles after 60 min of milling time and the transition to a glassy state with isotropy of properties by the end of the tribochemistry cycle. The dynamic properties of lacosamide were established: elastic and plastic deformation boundaries, region of inhomogeneous deformation and fracture point. The ratio of dissolution rate constants in water of samples before and after a full cycle of loading was 2.4 times. The lacosamide sample, which underwent a full cycle of mechanical loading, showed improved kinetics of API release by analysing dissolution profiles in 0.1 M HCl medium. The observed activation energy values of the cell death biosensor process in aqueous solutions of the lacosamide samples before and after the complete tribochemical cycle are were: 207 kJmol-1 and 145 kJmol-1, respectively. The equilibrium time of dissolution and activation of cell biosensor death corresponding to 20 minutes of mechanical loading on a solid was determined. The current conducted study may have an important practical significance in the transformation and management of drug substances properties of drug substances in solid form and in solutions, in increasing the strength of drug matrices by pre-strain hardening due to structural rearrangements during mechanical loading.

Point 2: Please focus discussing the objective of the current research in the introduction section of the manuscript.

Response 2: Thank you, Totally agree with the comment.

Information on the lacosamide pharmacodynamics and pharmacokinetics has been added to the Introduction section prior to the statement of study aim.

Point 3: Is the process of knife grinding a common practice for side reduction of active pharmaceutical ingredients?

Response 3: Thank you for this question!

Mechanical loading of a solid accompanied by grinding is aimed at its activation as a result of the accumulation of structural point defects, or solid-phase chemical transformations. The absence of solvents and by-products of the reaction, as well as the increase in the reaction rate due to the increase in the number and area of contacts, give significant advantages to this method, which belongs to the ‘green’ chemistry.

Point 4: Within the results, please discuss the results comparing the characteristics of lacosamide before and after milling.

Response 4: Thank you very much!

Each of the graphs in the Results section demonstrates the properties of the solid substance of lacosamide or its solutions compared before and after mechanical loading was carried out in the knife mill.

Point 5: Please improve the discussion of dissolution studies.

Response 5: Thank you.

The section “In vitro equivalence dissolution test (CDKT)” has been revised.

With gratitude, the authors
